# Predicting Voluntary Exercise Training among Korean Firefighters: Using Elicitation Study and the Theory of Planned Behavior

**DOI:** 10.3390/ijerph17020467

**Published:** 2020-01-10

**Authors:** Chung Gun Lee, Susan E. Middlestadt, Seiyeong Park, Junhye Kwon, Kyoungmin Noh, Dong-il Seo, Wook Song, Jung-jun Park, Han-joon Lee, Hyun Joo Kang, Yeon Soon Ahn

**Affiliations:** 1Department of Physical Education, College of Education, 71-1, Seoul National University, Seoul 08826, Korea; pseiy09@snu.ac.kr (S.P.); 2017_25059@snu.ac.kr (J.K.); nkm0909@snu.ac.kr (K.N.); songw3@snu.ac.kr (W.S.); 2Institute of Sport Science, Seoul National University, Seoul 08826, Korea; 3Department of Applied Health Science, School of Public Health, Indiana University, Bloomington, IN 47405, USA; semiddle@indiana.edu; 4Department of Sport Science, College of Liberal Arts, Dongguk University, Gyeongju 38066, Korea; seodi74@dongguk.ac.kr; 5Institute on Aging, Seoul National University, Seoul 08826, Korea; 6School of Sport Science, Pusan National University, Pusan 46241, Korea; jjparkpnu@pusan.ac.kr; 7School of Sport Science, Ulsan University, Ulsan 44610, Korea; hanjoon@ulsan.ac.kr; 8Department of Sport Medicine, College of Natural Science, Soonchunhyang University, Asan 31538, Korea; violethjk@naver.com; 9Department of Preventive Medicine and Genomic Cohort Institute, Yonsei Wonju College of Medicine, Yonsei University, Wonju 26493, Korea; ysahn1203@gmail.com

**Keywords:** firefighter, exercise training, salient belief

## Abstract

*Background*: Firefighters are required to have high levels of aerobic and anaerobic power because they often perform physically demanding work in dangerous environments. Therefore, it is important to find out salient factors influencing voluntary exercise training among Korean firefighters based on well-validated theory. *Methods*: The present study conducted an elicitation study to elicit salient behavioral, normative, and control beliefs about exercise training among Korean firefighters and identified salient beliefs that have a significant indirect effect on behavior through intention using structural equation modeling. *Results*: Although ten modal salient beliefs obtained from our elicitation study are similar to those elicited from previous TPB belief-based research with a focus on exercise behavior, only three of these (i.e., “improves my physical ability” (coef. = 0.078, *p* = 0.006), “takes too much time” (coef. = 0.064, *p* = 0.023), and “colleagues” (coef. = 0.069, *p* = 0.016) indirectly influenced exercise training behavior through intention among Korean firefighters. *Conclusions:* Our results may contribute to the literature by providing important information suggesting that three modal salient beliefs are major cognitive determinants of exercise training behavior among Korean firefighters and they may play an essential role in developing effective programs or policies for promoting Korean firefighters’ exercise training.

## 1. Introduction

Firefighters often perform physically demanding works in dangerous environments. They must use and carry heavy tools, climb ladders and stairs, and they may also be called upon to perform difficult and dangerous rescue operations. Further, these works must be done with time urgency while they encounter toxic smoke, extreme temperatures, and chaotic conditions that include low visibility and loud noise. In addition, firefighters must wear personal protective equipment when they perform their work. Although personal protective equipment is necessary to protect firefighters, it also imposes a substantial physiological burden because of its restrictiveness, weight, and insulative properties. Therefore, in order to perform firefighting tasks properly, firefighters are required to have high levels of aerobic and anaerobic power [1,2]. Although firefighters’ anaerobic power is rarely investigated [3,4,5,6] and no minimum limits of anaerobic power have ever been suggested, maximum aerobic power (VO_2max_) of 42 ml/min/kg is recommended by the National Fire Protection Association (NFPA) [7]. However, in 2018, the average VO_2max_ of Korean firefighters was 40.23 ml/min/kg and only 44.97% of Korean firefighters were performing at least 150 minutes of moderate- to vigorous-intensity exercise training per week [8]. This is a serious problem because low participation in exercise training results in poor physical fitness which, in turn, has been strongly associated with decreased job performance during firefighting activities [9,10,11]. A legislation of the Korean National Fire Agency (NFA; e.g., a physical fitness test contributes only 5 percent of the total score of the promotion examination) may have contributed to low participation in exercise training and poor physical fitness among Korean firefighters [12] but it is also important to find out salient factors influencing voluntary exercise training (i.e., participating in exercise training of one’s own free will) among Korean firefighters based on well-validated theory.

The theory of planned behavior (TPB) is a well-validated and widely used theoretical framework for explaining health-related behaviors, including exercise [13]. As shown in Figure 1, the TPB posits that certain behavior is formulated by intention to perform that behavior, which, in turn, is strongly affected by subjective norm (social approval or disapproval) and attitude (positive or negative evaluation) toward that behavior and perceived behavioral control (ease or difficulty) over performance of that behavior [14]. Moreover, the TPB also posits that attitude, subjective norm, and perceived behavioral control are determined by behavioral (beliefs about the advantages or disadvantages of performing the behavior and the evaluations of those outcomes), normative (beliefs about whether others approve or disapprove of performing the behavior and motivation to comply with those expectations), and control beliefs (beliefs about the presence of facilitators or constraints of the behavior and the perceived power of those factors), respectively [14]. Although people can hold many beliefs about certain behavior, they can utilize only a relatively small number of beliefs about that behavior at any given moment [14,15]. These salient beliefs are considered to be the predominating determinants of intention and behavior and play an essential role in devising effective behavior change interventions [14].

Although the TPB is specified at the individual level, in practice, it is more convenient to identify a set of the most commonly held beliefs in a given population [16]. An elicitation study can be conducted to identify these modal salient beliefs in a representative sample of the target population. To elicit modal salient beliefs in a given population, researchers should conduct an open-ended question interview for assessing the salient behavioral, normative, and control beliefs and perform a content analysis of open-ended responses to determine the 5 to 10 most frequently reported beliefs. Although an elicitation study is important because it provides researchers with a substantial amount of information about the behavior’s cognitive foundation in a given population [16], most exercise studies that used the TPB are conducted without it [17]. The TPB exercise studies utilizing elicitation study are warranted for at least three reasons. First, it is important to find out the modal salient beliefs about exercise in diverse populations because not all populations share the same feelings and thoughts about exercise [18]. Second, most previous TPB exercise studies that had conducted an elicitation study could not determine whether participants in the main study and the elicitation study were similar. When researchers cannot obtain correspondence between the elicitation study and the main study participants, it is possible that the main study may have been examined people who have different salient beliefs about exercise than the elicitation study [18]. Third, since not all modal salient beliefs affect intention and behavior, researchers should identify modal salient beliefs that significantly affect behavior through intention using statistical methods such as structural equation modeling. Although this procedure enables researchers to obtain more detailed information about the cognitive foundation of behavior and to develop more effective behavior change interventions based on the obtained information, to our knowledge, there were only two TPB exercise studies that used such an approach [19,20]. The present study, therefore, aims to conduct an elicitation study to elicit salient behavioral, normative, and control beliefs about exercise training among Korean fire fighters following guidelines for the elicitation study suggested by Ajzen and Fishbein (1980) and to identify salient beliefs that have significant indirect effect on behavior through intention using structural equation modeling. 

## 2. Materials and Methods 

### 2.1. Data

Prior to collecting data for the main study, an elicitation study was conducted in order to elicit modal salient beliefs (i.e., behavioral, normative, and control beliefs) among target firefighters following the procedure suggested by Fishbein and Ajzen [21]. A total of 29 firefighters were recruited to participate in a semi-structured interview that included open-ended questions regarding participation in voluntary exercise training. The think-aloud method was used to encourage firefighters to indicate as many factors as possible regarding voluntary exercise training under each TPB construct [22]. The elicitation process continued until saturation was reached. Content analysis was conducted for analyzing responses of participants, and the most frequently mentioned beliefs (i.e., beliefs that were mentioned by more than 30% of the total participants) were used as a basis for developing a standard questionnaire that is used in the main study. 

Total 175 firefighters (159 males and 16 females) were recruited for the main study from a fire station located in Seoul, South Korea. Only firefighters whose main job is to suppress fire, rescue people in danger, or give first aid to victims were recruited because their work is more crucially related to physical ability compare to firefighters who spend most of their time in the office (e.g., administrative staff). The head of the fire station provided permission for employee participation, and administrative staff agreed to assist trained investigators in conducting a survey during their working hours. The principle investigator trained graduate students to conduct a survey and to present the opportunity to participate in the survey to all firefighters working in the target fire station. The graduate students also distributed consent forms to target firefighters. Among the total of 215 firefighters working in the target fire station, 175 (81.4%) firefighters provided written assent for completing the survey. Participating firefighters were then given a packet that included an incentive worth $5.00, letters of support, and survey questionnaires. During the survey, the graduate students remained in the fire station and responded to any questions posed by participating firefighters. To measure the participants’ voluntary exercise training during the following month, a one-month follow-up survey was also conducted (the response rate was 97.7%, *N* = 171). All survey data were collected in August 2018 because we wanted to minimize the influence of the annual physical fitness test that was conducted on April 2018 for all firefighters in South Korea. This study was approved by Yonsei University Institutional Review Board (IRB No. CR318031).

### 2.2. Measures

In an elicitation study, participants were asked to indicate the advantages and disadvantages of participating in exercise training (behavioral beliefs). The behavioral beliefs that were mentioned by more than 30% of the participants (nine participants) were “improves my physical ability”, “makes me tired”, “causes injury”, and “takes too much time”. Participants were also asked to indicate any individuals or groups who would approve or disapprove of their exercise training (normative beliefs). The normative beliefs that were mentioned by more than nine participants were “family members” and “colleagues”. Participants were asked to identify factors that may impede or facilitate their participation in exercise training (control beliefs). The control beliefs that were mentioned by more than 30% of the participants were “exercise facilities”, “exercise partners”, “lack of time”, and “not in good physical condition”. All these elicited beliefs (see Table 1) were used as a basis for the development of a main study questionnaire.

Behavioral belief strength was measured using four behavioral beliefs elicited from the elicitation study. Participants were asked to rate how likely the benefits (e.g., “improves my physical ability”) and costs (e.g., “makes me tired”) would result when they participate in exercise training in the forthcoming month. A 7-point Likert scale (−3 = extremely unlikely to 3 = extremely likely) was used to assess these items. The outcome evaluation corresponding to each behavioral belief strength was also measured on a 7-point Likert scale (−3 = extremely bad to 3 = extremely good). The strength of each behavioral belief was multiplied by corresponding outcome evaluation to represent one aspect of attitude toward participating in exercise training.

The strength of normative belief was measured using two normative beliefs obtained in the elicitation study. Participants were asked to answer how likely their family members and colleagues think that they should participate in exercise training in the forthcoming month. A 7-point Likert scale (−3 = extremely disagree to 3 = extremely agree) was used to measure these two items. The motivation to comply with each referent was also assessed on a 7-point Likert scale (0 = not at all to 6 = very much). The strength of each normative belief was then multiplied by corresponding motivation to comply to represent one aspect of subjective norm toward participation in exercise training.

Control belief strength was assessed using four kinds of control beliefs elicited from the elicitation study. Participants were requested to rate how likely they would be confronted with difficult (e.g., “lack of time”) and favorable (e.g., “exercise partners”) situations when they participate in exercise training during the next month. To assess control belief strength, a 7-point Likert scale (0 = extremely unlikely to 6 = extremely likely) was used. The perceived ability to handle the situation (perceived power) corresponding to each control belief strength was also assessed on a 7-point Likert scale (−3 = extremely difficult to 3 = extremely easy). Each perceived power was multiplied by corresponding control belief strength to represent one aspect of perceived behavioral control over participating in exercise training.

Intention to participate in exercise training in the forthcoming month was measured using three questions (e.g., “How likely is it that you will try to participate in exercise training within the next month?”). These three items were rated on a 7-point Likert scale (bipolar) ranging from extremely unlikely to extremely likely. The mean score for these three items was calculated to represent an intention to participate in exercise training in the next month.

In the one-month follow-up survey, two questions were used to assess the level of exercise training from the previous month. Participants were asked for the number of days they performed exercise training per week during the past month. They were also asked to answer how many total minutes they spent doing exercise training on a day of exercise. The number of days exercised per week was multiplied by total daily minutes spent exercising to calculate the total minutes spent on exercise training per week in the past month.

### 2.3. Statistical Analysis

The present study examines the effects of the modal salient beliefs on exercise training among Korean firefighters (*N* = 171). The structural equation modeling was used to test the expected direct and indirect effects of modal salient beliefs on intention to participate in exercise training and exercise training behavior, respectively. The structural equation model was developed based on the hypothesized theoretical framework shown in Figure 2. The parameters in our model were estimated using the full-information maximum likelihood (FIML) method. There were no missing values on variables. The Sobel method was used to examine indirect effects [23]. The goodness of fit was evaluated using chi-square, comparative fit index (CFI), standardized root mean square residual (SRMR), and root mean square error of approximation (RMSEA). It is suggested that a model fits the data well when chi-square value is insignificant, SRMR is lower than 0.05, CFI is higher than 0.95, and RMSEA is lower than 0.05. The Pearson correlation coefficient was used to examine correlations among exogenous variables. All the analyses were conducted using Mplus Version 7 (Muthen & Muthen, Los Angeles, CA, USA) [24].

## 3. Results

The descriptive statistics for study variables are presented in Table 1. Among the behavioral beliefs, “improves my physical ability” had the highest mean scores for both behavioral belief strength (mean = 2.440) and outcome evaluation (mean = 2.491), whereas the lowest mean scores for behavioral belief strength (mean = −0.149) and outcome evaluation (mean = −2.411) were observed in “takes too much time” and “causes injury”, respectively. Both “family members” and “colleagues” had similar mean scores for normative belief strength and motivation to comply. Among the control beliefs, the lowest mean scores for control belief strength (mean = 2.606) and perceived power (mean = −0.623) were observed in “not in good physical condition” whereas “exercise partners” had the highest mean scores for both control belief strength (mean = 3.514) and outcome evaluation (mean = 1.669). The mean score for intention to participate in exercise training was 1.939 and the mean total minutes spent on exercise training per week was 210.614 minutes. The correlations among exogenous variables ranged from −0.15 to 0.47 (see Table 2).

Fit indices suggested that our theoretical model fits the data well (χ^2^ = 10.193 [df = 10, *p* = 0.424 > 0.05], SRMR was 0.039 < 0.05, CFI = 0.997 > 0.95, and RMSEA = 0.011 < 0.05). As shown in Figure 2, only three modal salient beliefs had a significant effect on intention to participate in exercise training. The more participants think that doing exercise training improves their physical ability and the improvement of physical ability is good, the more likely participants are to have higher intention to do exercise training (β = 0.254, standard error = 0.070, *p* < 0.01). The more firefighters think that participating in exercise training does not take too much time and taking too much time is bad, the more likely firefighters are to have higher intention to do exercise training (β = 0.209, standard error = 0.078, *p* < 0.01). The more participants believe that their colleagues think they should do exercise training and they want to comply with their colleagues, the more likely participants are to have higher intention to do exercise training (β = 0.226, standard error = 0.077, *p* < 0.01). The exercise training behavior was also significantly and strongly affected by intention to exercise (β = 0.308, standard error = 0.069, *p* < 0.01). In addition, the model explained 25% of the variance in intention to participate in exercise training and 10% of the variance in exercise training behavior.

The indirect effects from three modal salient beliefs (i.e., “improves my physical ability”, “takes too much time”, and “colleagues”) to exercise training behavior were also examined (Table 3). “Improves my physical ability” had the strongest indirect effect on exercise training through intention (β = 0.078, standard error = 0.028, *p* < 0.01), followed by “colleagues” (β = 0.069, standard error = 0.029, *p* < 0.05), and “takes too much time” (β = 0.064, standard error = 0.028, *p* < 0.05).

## 4. Discussion

The findings of the present study confirm that the TPB is a useful framework to understand voluntary exercise training behavior among Korean firefighters. The structural equation modeling revealed an excellent fit between our theoretical model and the data. Identification of modal salient beliefs that significantly influence behavior through intention is important because such procedure is vital to the formation of efficient interventions for changing behavior. Although ten modal salient beliefs obtained from our elicitation study are similar to those elicited from previous TPB belief-based research with a focus on exercise behavior [18], only three of these (i.e., “improves my physical ability”, “takes too much time”, and “colleagues”) indirectly influenced exercise training behavior through intention among Korean firefighters. 

The more Korean firefighters believe that performing exercise training improves their physical ability and the improvement of physical ability is a good thing, the more likely they are to participate in exercise training. “Improves physical ability” is an occasionally identified salient exercise belief, especially among undergraduate students [18]. However, its effect on exercise behavior and intention has not yet been tested in previous studies. The significant effect of this belief on exercise training behavior among Korean firefighters may indicate that physical ability is closely related to Korean firefighters’ job performances. Although a physical fitness test contributes only 5 percent of the total score of a firefighter promotion examination in South Korea [12], Korean firefighters need to have high levels of physical ability so that they can rescue people in difficult situations. They may want to have a stronger body to protect themselves and to rescue others efficiently. However, health and fitness coordinators who work in fire stations located in South Korea do not currently have sufficient professional knowledge about exercise training [25]. A scientifically proven exercise training program for enhancing aerobic and anaerobic power of firefighters, therefore, needs to be developed and introduced to Korean firefighters to promote their voluntary exercise training.

Another behavioral belief that significantly affected voluntary exercise training through intention is “takes too much time”. “Takes too much time” is one of the most frequently reported salient exercise beliefs in various populations, including worksite employees [18]. Cowie and Hamilton also found that this belief significantly affects both exercise behavior and intention among university students [20]. According to the Korea Labor Institute (KLI), approximately 14 percent of Korean firefighters reported that they do not have enough time to exercise due to the heavy workload [26]. In addition, “not enough time to do exercise training” was one of the most commonly reported issues among Korean firefighters in previous research [25,27]. These results suggest that some managerial and institutional barriers exist when Korean firefighters are trying to participate in exercise training. To resolve this problem, the upper echelons need to reduce the unnecessary workload of firefighters and provide conditions that allow firefighters to do exercise training during their work hours.

One normative belief (i.e., “colleagues”) appears to have a significant indirect effect on voluntary exercise training through intention among Korean firefighters. “Colleagues” is also a frequently identified salient normative referent in previous elicitation research on exercise behavior, especially among worksite employees [18]. Utilizing wearable health devices can be an efficient way to promote interaction among firefighters in the same fire station and this social interaction may enhance firefighters’ participation in exercise training by group dynamics [28], having common exercise goals, exchanging experiences, or competing against each other [29]. As a result, firefighters may reflect more on their exercise training due to an increase in the belief that their colleagues think that they should participate in exercise training.

The present study is not without limitations. First, the results of this study may not be generalizable to all Korean firefighters because this study only included a single fire station in South Korea. Second, the use of self-report questionnaires when measuring exercise training could result in social desirability bias, recall bias, or response bias. It would have been beneficial to use objectively measured exercise training in order to improve accuracy of results [30]. Third, our TPB model explained only 25% of the variance in intention to participate in exercise training and 10% of the variance in exercise training behavior. This is partly because our model only focused on the cognitive foundation of exercise training behavior. The interplay between cognitive factors and environmental constraints in determining exercise training behavior, therefore, needs to be considered when developing interventions for promoting Korean firefighters’ exercise training based on the results of this study [31]. Fourth, since there are several different types of exercise training, more specific information regarding exercise training is needed to distinguish at least aerobic exercise training from anaerobic exercise training. To better understand the influence of salient beliefs on exercise training among firefighters, future studies need to utilize a more precise measure of exercise training.

## 5. Conclusions

Despite these limitations, this study is the first study that incorporates individual salient beliefs to test the full TPB model in the exercise domain. Our results may contribute to the literature by providing important information, suggesting that three modal salient beliefs (i.e., “improves my physical ability”, “takes too much time”, and “colleagues”) are major cognitive determinants of exercise training behavior among Korean firefighters and they may play an essential role in developing effective programs or policies for promoting Korean firefighters’ exercise training.

## Figures and Tables

**Figure 1 ijerph-17-00467-f001:**
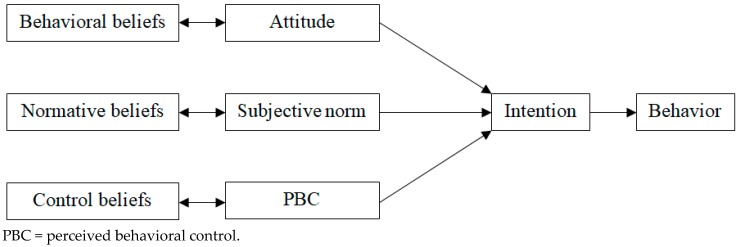
The theory of planned behavior [14].

**Figure 2 ijerph-17-00467-f002:**
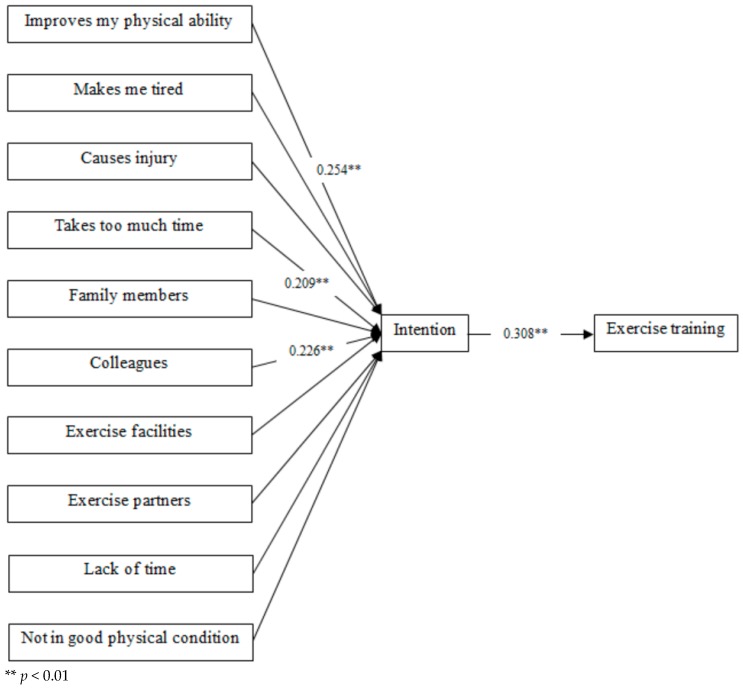
Structural equation modeling (*N* = 171).

**Table 1 ijerph-17-00467-t001:** Descriptive statistics for participants (*N* = 175).

Variables		Mean	(SD)	Maximum	Minimum
Improves my physical ability	BB	2.440	(0.621)	−1.000	3.000
	OE	2.491	(0.576)	−1.000	3.000
Makes me tired	BB	0.634	(1.555)	−3.000	3.000
	OE	−0.897	(1.668)	−3.000	3.000
Causes injury	BB	0.297	(1.623)	−3.000	3.000
	OE	−2.411	(1.035)	−3.000	2.000
Takes too much time	BB	−0.149	(1.640)	−3.000	3.000
	OE	−1.629	(1.122)	−3.000	2.000
Family members	NB	2.126	(0.770)	−2.000	3.000
	MC	4.526	(1.321)	0.000	6.000
Colleagues	NB	2.126	(0.593)	−1.000	3.000
	MC	4.697	(1.196)	0.000	6.000
Exercise facilities	CB	3.109	(1.753)	0.000	6.000
	PP	−0.371	(1.599)	−3.000	3.000
Exercise partners	CB	3.514	(1.678)	0.000	6.000
	PP	1.669	(1.090)	-3.000	3.000
Lack of time	CB	3.206	(1.645)	0.000	6.000
	PP	−0.594	(1.474)	−3.000	3.000
Not in good physical condition	CB	2.606	(1.608)	0.000	6.000
	PP	−0.623	(1.476)	−3.000	3.000
Intention		1.939	(1.091)	−3.000	3.000
Exercise training (minute/week)		210.614	(192.496)	0.000	1500.000

SD = standard deviation, BB = behavioral belief, OE = outcome evaluation, NB = normative belief,.MC = motivation to comply, CB = control belief, PP = perceived power.

**Table 2 ijerph-17-00467-t002:** Correlations among exogenous variables (*N* = 175).

	1	2	3	4	5	6	7	8	9	10
1. Improves my physical ability										
2. Makes me tired	0.08									
3. Causes injury	−0.02	0.47								
4. Takes too much time	0.09	0.31	0.38							
5. Family members	0.27	0.25	0.32	0.15						
6. Colleagues	0.26	0.21	0.17	0.13	0.50					
7. Exercise facilities	−0.09	0.11	0.19	0.08	0.01	0.02				
8. Exercise partners	0.15	0.20	0.27	0.42	0.23	0.22	0.16			
9. Lack of time	−0.15	0.25	0.15	0.24	−0.05	−0.06	0.21	0.19		
10. Not in good physical condition	−0.09	0.34	0.27	0.22	0.01	0.07	0.17	0.13	0.33	

**Table 3 ijerph-17-00467-t003:** Indirect effects of modal salient beliefs on participation in exercise training (*N* = 171).

Indirect effects		Coefficients	(SE)
From	To		
Improves my physical ability	Exercise training	0.078	(0.028) **
Takes too much time	Exercise training	0.064	(0.028) *
Colleagues	Exercise training	0.069	(0.029) *

SE = standard error; * *p* < 0.05, ** *p* < 0.01.

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
