# Peer review of "Predicting Voluntary Exercise Training among Korean Firefighters: Using Elicitation Study and the Theory of Planned Behavior"

_ijerph, 2020, doi:10.3390/ijerph17020467_

Round 1

Reviewer 1 Report

Very good and interesting topic.

For improving:

to correct abstract- three determinants put in to the results not in the conclusions comment the possibility of influence the behavior with this three determinants did you do a correlation between determinants and realy cardiovascular fitness??

Reviewer 2 Report

Thanks for inviting me as one reviewer to judge this study. Overall, this study is of particular significance owing to its population. However, some minor concerns of this study should be mentioned for further consideration.

Line 22: ‘works’ should be replaced with ‘work’; also see Line 39;

Line 44: personal protective equipment (PPE) appears once across the manuscript, so the abbreviation should not be used;

Line 51: removing the ‘moreover’;

Line 107: ‘total’ should be replaced with ‘a total of’;

Line 105-114: In method section, it seems you develop a new survey questionnaire based on an in-advance investigation. Thus, how is the psychrometries of the newly developed questionnaire, such as internal consistency, reliability or validity?

Line 133: Do you have sufficient evidence to support your validation of measures applied in this study?

In discussion section, I recommend that you may add some content of comparing with other similar study findings. What is your study’s clinical relevance? I believe that adding some discussion regarding this study’s practical implications would be better. Overall, the discussion section should be re-organized and re-presented. There is a structure of discussion section that you could follow”

-Main findings of this study

-Comparisons with other studies

-Using TPB to explain your major research findings

-Providing practical implications for research and practice

-Strengths and limitations of this study

Reviewer 3 Report

General comment

The authors provided us with data regarding the Korean firefighter’s beliefs about factors that determine voluntary participation in exercise training. Here, they show some important aspects that may lead to a higher or a reduced involvement in training. However, at some point of the intervention, this written factors are interpreted as a belief from one firefighter that is interpreting other beliefs. With this in mind, I leave to the authors some comments for consideration.

Specific comments

I don’t think da VO2max can be used as abbreviation from minimum aerobic capacity. Within the scientific community, VO2max is known as maximum oxygen uptake from a single subject. Please revise.

Why authors state that 150min from moderate to vigorous exercise training is a problem for the firefighter’s capacity? Can you show us the thresholds that are desirable for this kind of exercise training?

How your TPB impacts on aerobic and anaerobic capacity of the firefighters? This rational should be better explained in your introduction section. I think that here you should better clarify your research gap.

What do you mean with “voluntary participation”. Please describe.

Authors should try to convince me why this question is important for someone (e.g. firefighter) that has a job and needs to perform that job effectively. I ask this because the previous studies that used TPB related to exercise, were linked to a “participation context”, where the involvement in exercise was dependent on the subject will and where he/her was not being paid for that. The firefighter’s case is so distinct that needs that clarification.

Why the authors restricted the survey just to one fire station? Why not expand to others that may have different approaches to exercise training?

I saw “takes to much time” and “lack of time” in behavioral and control beliefs, respectively. From my point of view this is the same.

Why authors opted to characterize the firefighters training by multiplying the number of days by the total minutes? Why not interpreted the training regime in terms of aerobic and anaerobic training zone?

Your table 2 show us some correlation values between variables, but no correlation treatment was given in the statistical analysis section. Please rectify.

You refer that your theoretical model fits the data well. But, at any time (.e.g at the statistics section) I see any thresholds for this kind of interpretation, if it is well, good, poor…etc.

At some point your results lead to us interpreting what some subjects think that the other are thinking. For me this kind of analysis is somehow disturbing. It something like “I’m thinking in what the other person is thinking about the think from the third person”.

Probably, more that to test beliefs about training participation, you should put your effort in understanding what kind of factors the firefighters think that should better be used and trained in each session.

You model represented in your figure 2 probably should give to us how differently (increase intention or decrease intention) about participation in training. I ask this because “improves my physical ability” and “takes too much time” are contradicting statements that show a positive and significant value, but should be interpreted differently.

Your discussion is what your results express. I’ve no comments at this stage for this section.

Round 2

Reviewer 2 Report

After detailed responses and thorough revisions throughout the manuscript, I think this study is qualified for publication.

Author Response

After detailed responses and thorough revisions throughout the manuscript, I think this study is qualified for publication.

--> Thank you again for reviewing our manuscript.

Reviewer 3 Report

General comment

The authors made smooth changes considering the reviewer opinion. I still have some doubts that remain for a better understanding.

Specific comments

42 ml/kg/min is a measure of maximal oxygen uptake (VO2max). Considering the aerobic energy sources, the VO2max is known as the aerobic “power” source and not as the aerobic “capacity” source. Please revise.

Once again I ask for the authors to better highlight in their introduction section how the TPB method impacts on aerobic and anaerobic capacity of the firefighters. This makes sense to gather the both ideas of your paragraphs. Without this your information about the aerobic capacity of the firefighters adds nothing to your introduction.

I maintain my previous comment! Why authors opted to characterize the firefighters training by multiplying the number of days by the total minutes? Why not interpreted the training regime in terms of aerobic and anaerobic training zone? I can train a higher number of ours and not train in the training zone that is supposed to. From my perception, a more detailed info is needed.

I acknowledge the authors effort in adding some statistical data to the model fitting. But what I’m asking is that in the “statistics section” they should add the values for fitting interpretation, if it is well, good, poor…etc.

Considering your explanation for your figure 2, and seeing that the response “takes too much time” has a positive coefficient of 0.209, this lead us to think that, how bigger  will be the training duration, higher will be the intention to go for it. For my perception this approach is wrong and should be revised. Moreover, this is contrary to what is observed in university students and should be better discussed.
